**Data Availability Statement:** Data cannot be shared publicly because clinical data is confidential

# Three year clinical outcomes in a nationwide, observational, siteless clinical trial of atrial fibrillation screening—mHealth Screening to Prevent Strokes (mSToPS)

**Steven R. Steinhubl**[1]*, **Jill Waalen**[1], **Anirudh Sanyal**[2], **Alison M. Edwards**[2], **Lauren M. Ariniello**[1], **Gail S. Ebner**[1], **Katie Baca-Motes**[1], **Robert A. Zambon**[3], **Troy Sarich**[4], **Eric J. Topol**[1]

**1** Scripps Research Translational Institute, La Jolla, CA, United States of America, **2** Healthagen, LLC, Chicago, IL, United States of America, **3** Janssen Research and Development, Titusville, NJ, United States of America, **4** Johnson & Johnson, New Brunswick, NJ, United States of America

* steinhub@scripps.edu

## Abstract

### Background

Atrial fibrillation (AF) is common, often without symptoms, and is an independent risk factor for mortality, stroke and heart failure. It is unknown if screening asymptomatic individuals for AF can improve clinical outcomes.

### Methods

mSToPS was a pragmatic, direct-to-participant trial that randomized individuals from a single US-wide health plan to either immediate or delayed screening using a continuous-recording ECG patch to be worn for two weeks and 2 occasions, ~3 months apart, to potentially detect undiagnosed AF. The 3-year outcomes component of the trial was designed to compare clinical outcomes in the combined cohort of 1718 individuals who underwent monitoring and 3371 matched observational controls. The prespecified primary outcome was the time to first event of the combined endpoint of death, stroke, systemic embolism, or myocardial infarction among individuals with a new AF diagnosis, which was hypothesized to be the same in the two cohorts but was not realized.

### Results

Over the 3 years following the initiation of screening (mean follow-up 29 months), AF was newly diagnosed in 11.4% (n = 196) of screened participants versus 7.7% (n = 261) of observational controls (p<0.01). Among the screened cohort with incident AF, one-third were diagnosed through screening. For all individuals whose AF was first diagnosed clinically, a clinical event was common in the 4 weeks surrounding that diagnosis: 6.6% experienced a stroke,10.2% were newly diagnosed with heart failure, 9.2% had a myocardial infarction, and 1.5% systemic emboli. Cumulatively, 42.9% were hospitalized. For those

health plan data. Data are available from the Scripps Office for the Protection of Research Subjects(contact via Holmes. Jennifer@scrippshealth.org) for researchers who meet the criteria for access to confidential data.

**Funding:** Janssen Pharmaceuticals (SRS, GE, LA, KBM, AE, AS). https://www.janssen.com/us/ National Institutes of Health (NIH)/National Center for Advancing Translational Sciences UL1TR002550 (SRS, EJT, GE, LA) https://ncats. nih.gov Qualcomm Foundation (SRS, EJT, GE). The primary funder collaborated in the design and conduct of the study; preparation, review, or approval of the manuscript.

**Competing interests:** I have read the journal's policy and the authors of this manuscript have the following competing interests: Steve Steinhubl – NIH UL1TR002550 grant support; Qualcomm Foundation. Jill Waalen – No conflicts of interest Anirudh Sanyal - Employee, Healthagen, LLC Alison Edwards – Employee, Healthagen, LLC Lauren Ariniello - NIH UL1TR002550 grant support Gail Ebner - NIH UL1TR002550 grant support; Qualcomm Foundation Katie Baca-Motes - No conflicts of interest Bob Zambon- Employee, Janssen Pharmaceuticals (a subsidiary of Johnson & Johnson); Stockholder Johnson & Johnson. Troy Sarich – Employee, Johnson & Johnson; Stockholder Johnson & Johnson. Eric Topol – NIH UL1TR002550 grant support; Qualcomm Foundation This support does not alter our adherence to PLOS ONE policies on sharing data and materials.

diagnosed via screening, none experienced a stroke, myocardial infarction or systemic emboli in the period surrounding their AF diagnosis, and only 1 person (2.3%) had a new diagnosis of heart failure. Incidence rate of the prespecified combined primary endpoint was 3.6 per 100 person-years among the actively monitored cohort and 4.5 per 100 person-years in the observational controls.

## Conclusions

At 3 years, screening for AF was associated with a lower rate of clinical events and improved outcomes relative to a matched cohort, although the influence of earlier diagnosis of AF via screening on this finding is unclear. These observational data, including the high event rate surrounding a new clinical diagnosis of AF, support the need for randomized trials to determine whether screening for AF will yield a meaningful protection from strokes and other clinical events.

## Trail registration

The mHealth Screening To Prevent Strokes (mSToPS) Trial is registered on ClinicalTrials. gov with the identifier NCT02506244.

## Introduction

Atrial fibrillation (AF) is common, with a lifetime risk of nearly 40% for adults over age 55 [1]. It is estimated that nearly 38 million individuals worldwide have a diagnosis of AF [2], although this underestimates the true prevalence as studies have suggested that 13–30% of all individuals with AF are undiagnosed [3, 4]. While stroke prevention is a major emphasis of treatment once AF is identified, AF is also an independent risk factor for mortality, ischemic heart disease, heart failure and other significant morbidities [5]. In fact, in the 5 years following a diagnosis of AF, death is the most frequent clinical event in the years following diagnosis, followed by heart failure and then stroke [6]. For many with AF the diagnosis is first made at the time of presentation with an irreversible event including stroke [7, 8] and heart failure [9].

Because an ECG provides a definitive method for diagnosing AF, and the ability to perform one is rapidly expanding outside the healthcare setting to the individual, this has created unique opportunities to implement novel methods of screening for undiagnosed AF. However, currently, only opportunistic screening–screening of some people when circumstances allow via a pulse check or single ECG—is recommended by most professional society guidelines [10]. The 2020 European Guidelines give a IIa recommendation (weight of evidence/opinion is in favor of usefulness/efficacy) for systematic screening of individuals ≥75 years or those at high stroke risk [11].

To explore a novel program for AF screening in undiagnosed individuals, the mHealth Screening To Prevent Strokes (mSToPS) trial was carried out among members of a large, nationwide health insurance plan, incorporating a self-applied, wearable ECG patch [12]. In the primary analysis, enrolled participants were randomized to either immediate monitoring or delayed by 4 months, with immediate monitoring resulting in a nearly 9-fold greater incidence of new AF identification relative to routine care [12]. All participants who underwent monitoring (per protocol cohort) were then pooled and followed longitudinally, relative to matched observational controls, in order to explore the impact of screening on healthcare

resource utilization and clinical outcomes at 3 years. The one-year healthcare resource utilization results have been previously reported [13]. Here, we present the 3-year clinical outcomes. The original analysis plan to compare outcomes between individuals newly diagnosed with AF in the actively monitored and routine care cohorts was predicated on the assumption that AF rates would be similar at 3 years in both arms. However, as this proved to not be the case, in this report we will primarily focus on a descriptive analysis of the clinical outcomes.

## Methods

### Study design and oversight

The details of the design of the mSToPS trial have previously been described [12]. Briefly, it was an investigator-initiated, randomized (between immediate or delayed active monitoring), pragmatic trial involving Aetna health insurance members throughout the United States. The current analysis is the observational component of that study, with the 2 randomized arms combined into one actively monitored cohort, and their outcomes presented along with those of their matched, observational controls. The study was approved by the Scripps Office for the Protection of Research Subjects.

### Participant population

Inclusion criteria included age of ≥75 years, or a male over age 55 years or a female over 65 years with one or more co-morbidities. Exclusion criteria were a current or prior diagnosis of AF, atrial flutter, or atrial tachycardia, currently prescribed anticoagulation therapy, or having an implantable pacemaker and/or defibrillator.

As detailed in Fig 1, of an estimated ~360,000 eligible Aetna members, ~100,000 were randomly selected and sent information about the study, with the primary mode of enrollment being via email outreach. Full details of outreach and enrollment have been previously published [14]. 2659 individuals were enrolled between November of 2015 and October of 2016 and were all offered active monitoring, being randomized to monitoring commencing immediately or delayed by 4 months. Of these 2659 individuals 1738 (65.4%) participated in active monitoring. As the time frame of the current analysis begins on the day of monitoring initiation, and monitoring was delayed by 4 months in half of the monitored cohort, 20 monitored individuals are excluded from this analysis due to either a new clinical diagnosis of AF, a new excluded diagnosis (e.g. pacemaker) or having been disenrolled from the insurer prior to the initiation of monitoring, leaving 1718 actively monitored participants for long-term analysis.

For the routine-care concurrent observational cohort, 2 matched controls were selected for each of the actively monitored participants from the pool of individuals in the original eligible cohort who were not sent study trial outreach. An observational cohort was chosen due to the trial design precluding direct interaction with a healthcare provider and concerns for randomizing individuals to standard of care after receiving an outreach identifying them as being at potentially increased risk for AF without the opportunity to fully discuss the nuances of risk identification and their potential randomization to routine care. Matching was based on sex, age and $CHA_2DS_2$-VASc score. One-hundred five members of the control cohort were excluded from this analysis for reasons noted for the monitored cohort, leaving a final observation cohort of 3371 individuals.

### Study procedures

ECG screening was carried out using the iRhythm Zio$_®$$^{XT}$, an FDA-approved, single-use, 14-day, ambulatory ECG monitoring skin adhesive patch that monitors and retains in memory

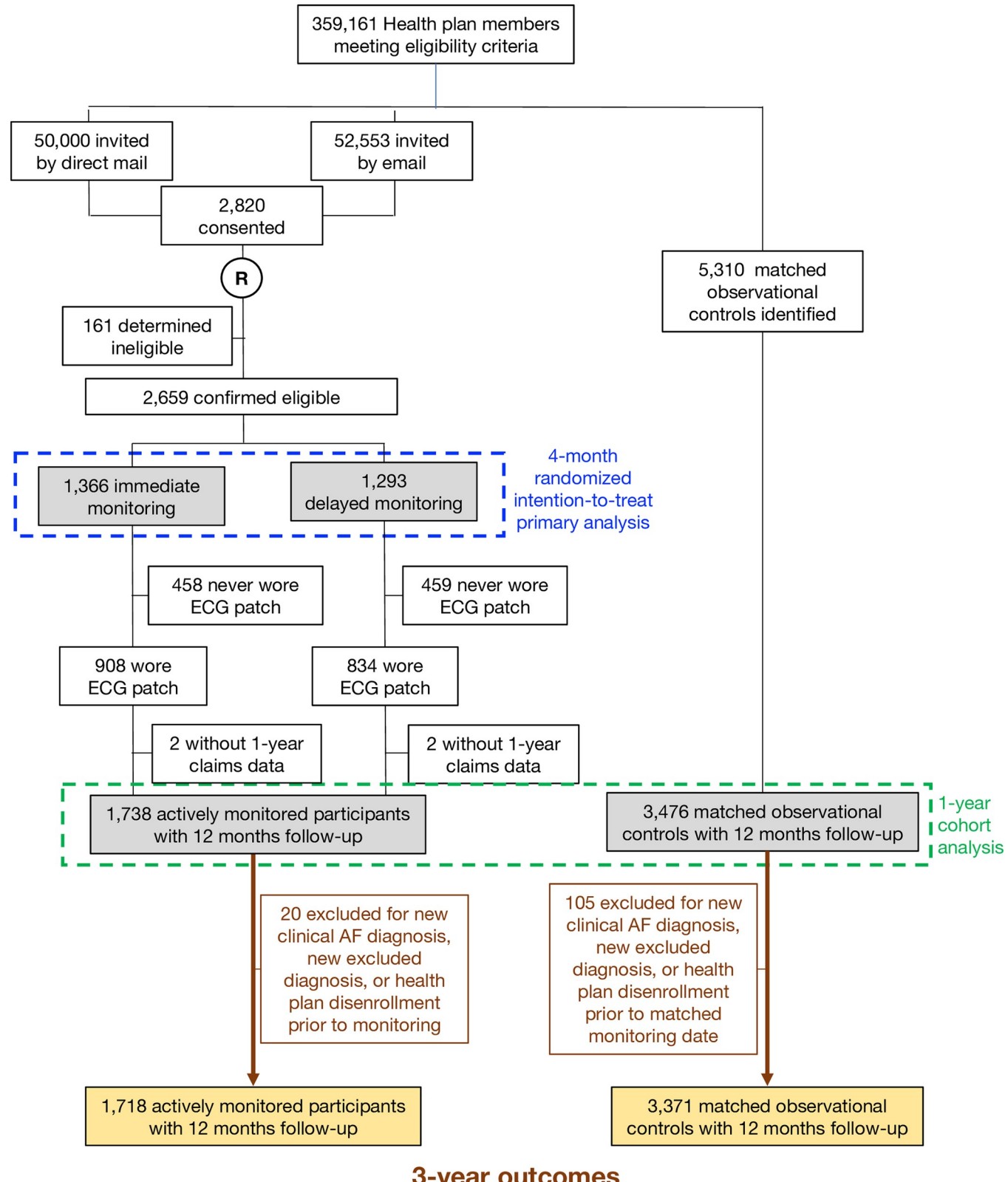

**Fig 1. CONSORT flow diagram.** Flow of participants beginning with all potentially eligible individuals, those enrolled and then those included in the 4 month, 1-year and 3-year analyses.

the wearer's continuous ECG. All participants were asked to wear 2 patches, one at the beginning and another 3-months later, with a median total duration of ECG monitoring of 24.7 days [12].

After each episode of active monitoring, if any potentially actionable results were identified, such as a finding of AF, or any sustained tachyarrhythmia, or prolonged pause, the participant was contacted by phone by the physician principal investigator to discuss the findings, which were then sent to the participant and, if they agreed, their physician, without associated treatment guidance. All participants received their full ECG patch results at the completion of the monitoring period.

## Study end points

The prespecified outcomes at 3 years after the initiation of ECG screening, determined under the assumption of similar AF rates in the monitored and control cohorts by that time, were the time to first event of the combined endpoint of stroke, systemic embolism, or myocardial infarction, determined by claims data, with, and without death, as determined by membership data in; A) individuals diagnosed with new AF in the 3 years following the initiation of screening, and B) the entire study cohort. Additional outcomes included the individual components of the combined endpoint. To explore the possible mechanism for our findings we examined the rate of diagnoses coinciding with or immediately proximate to an incident AF diagnosis. The primary safety endpoint was the incidence of hospitalizations with a primary diagnosis of bleeding. A new event was defined as a primary inpatient or ER diagnosis, or a primary outpatient diagnosis in an individual without a prior diagnosis at baseline. Diagnostic codes used to identify each clinical outcome are listed in the (S1 Checklist).

The start date for determination of outcomes from the Aetna database for both monitored participants and their matched controls was the date of initiating of monitoring. The end date was 36 months by exact day from the monitoring start date. Participants were censored at time of disenrollment from the health plan. Pharmacy data were missing for a subset of participants who were not enrolled in the insurer's pharmacy plan.

## Statistical analysis

Based on internal health plan data and available incidence data we anticipated an incidence of AF of 10% in both cohorts during the 3-year follow-up and an expected incidence of the combined endpoint of 12% in the control cohort and 5% among the actively monitored [15, 16], the sample size of 2,000 people in the per protocol actively monitored group and 4,000 matched controls yielded 81% power for a log rank test with a two-sided alpha = 0.05.

The primary outcome analysis (which can now only be considered exploratory because of the unanticipated finding of different AF rates in the monitored and control groups), was a comparison between the two cohorts of the time to event of the combined endpoint occurring within 3 years of screening initiation. Cumulative hazard curves were generated with the use of the Kaplan–Meier method. Univariable and multivariable analysis of time to the combined endpoint and each individual endpoint were performed using Cox proportional hazards models to include the following baseline (pre-randomization date) covariates: age, female sex, Charlson Comorbidity Index, heart failure, COPD, chronic renal failure, diabetes, hypertension, obesity, stroke, prior myocardial infarction, sleep apnea, baseline ER visits, baseline PCP visits, and baseline hospitalizations. The proportionality assumption was assessed by use of Schoenfeld residuals. Univariable and multivariable Poisson regression models were used to analyze total hospitalizations and hospitalizations for bleeding. Given the loose matching paradigm, a matching variable was not included in the primary analysis. No adjustment for

multiple comparisons was made, so conclusions regarding secondary analyses should be considered exploratory.

The software used for these analyses was SAS Enterprise Guide Version 6.1 (SAS Institute Inc., Cary, NC, USA).

# Results

## Study participants

A total of 1,718 actively monitored individuals and 3,371 observational controls were included in this analysis with a mean follow-up of 29 months. Mean (SD) age at enrollment was 73.8 years (7.0), 40.8% were female and median (IQR) baseline CHA2DS2-VASc was 3 (2–4). Baseline characteristics of the actively monitored and observational control cohorts are compared in Table 1 with imbalances in several co-morbidities.

## Diagnosis of atrial fibrillation

At the end of 3 years, AF was newly diagnosed in 11.4% (n = 196) of those actively monitored versus 7.7% (n = 261) in observational controls (p<0.01) (Fig 2A). In the actively monitored cohort, 65 individuals were first found to have AF through ECG patch screening and 131 were first diagnosed clinically. Over the 3-year follow-up, for those without AF on the patch, the rate of clinically diagnosed AF was 3.2 new diagnoses per 100 person-years, and 3.4 per 100 person-years for the observational cohort. During the last 18 months of follow-up, the difference in rate of new clinical diagnoses of AF between the 2 cohorts widened, with a lower rate in the monitored cohort that did not reach statistical significance (3.0 versus 4.0 per 100 person-years, p = 0.06) (Fig 2B).

**Table 1. Demographic characteristics at baseline of monitored individuals and matched controls.**

| | Entire Study Population | | Differences and 95% Confidence Intervals |
|---|---|---|---|
| | Actively Monitored Arm | Observational Control Arm | |
| | (n = 1718) | (n = 3371) | |
| Age (years), mean (SD) | 73.8 (7.0) | 73.7 (7.0) | (matched) |
| Female, n (%) | 699 (40.7) | 1374 (40.8) | (matched) |
| CHA$_2$DS$_2$ VASc Score, median (Q1-Q3) | 3 (2–4) | 3 (2–4) | (matched) |
| Charlson Comorbidity Index Score, mean (SD) | 5.3 (2.9) | 5.2 (2.8) | 0.05 (-0.12 to 0.22) |
| Stroke, n (%) | 218 (12.7) | 323 (9.6) | 3.1 (1.3 to 5.0) |
| Heart Failure, n (%) | 84 (4.9) | 196 (5.8) | -0.9 (-2.2 to 0.4) |
| Hypertension, n (%) | 1290 (75.1) | 2589 (76.8) | -1.7 (-4.2 to 0.8) |
| Diabetes Mellitus, n (%) | 598 (34.8) | 1195 (35.4) | -0.6 (-3.4 to 2.1) |
| Sleep Apnea, n (%) | 459 (26.7) | 699 (20.7) | 6.0 (3.5 to 8.5) |
| Prior Myocardial Infarction, n (%) | 91 (5.3) | 231 (6.9) | -1.6 (-2.9 to -0.2) |
| Chronic Obstructive Pulmonary Disease, n (%) | 137 (8.0) | 341 (10.1) | -2.1 (-3.8 to -0.5) |
| Obesity, n (%) | 288 (16.8) | 601 (17.8) | -1.1 (-3.3 to 1.1) |
| Chronic Renal Failure, n (%) | 182 (10.6) | 305 (9.0) | 1.6 (-0.2 to 3.3) |
| Primary care visits per person-year prior to enrollment | 2.64 | 2.64 | -0.006 (-0.14 to 0.13) |
| Cardiologist visits per person-year prior to enrollment | 0.67 | 0.50 | 0.16 (0.10 to 0.23) |
| Emergency Department visits per 100 person-years prior to enrollment | 13.15 | 18.21 | -5.06 (-5.38 to -4.74) |
| Hospitalizations per 100 person-years prior to enrollment | 5.70 | 6.47 | -0.76 (-0.96 to -0.56) |

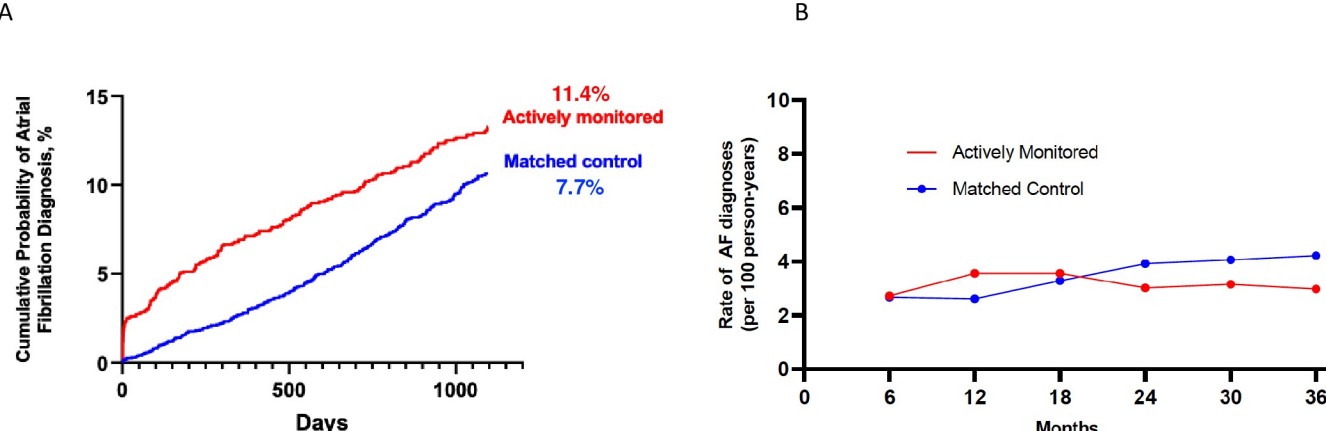

**Fig 2. Atrial fibrillation diagnosis.** A.) Cumulative Probability of a diagnosis of atrial fibrillation in the 3 years following the initiating of monitoring. B.) Rate of new diagnosis of atrial fibrillation in the monitored and observational cohorts after completion of active monitoring. All new diagnoses occurring annually after 6 months from the initiation of screening.

In the subset of individuals diagnosed with AF in whom pharmacy data was available (126 of 261 controls and 116 of 196 actively monitored), 45.2% and 44.0%, respectively, were initiated on an anticoagulant (p = 0.84).

## Prespecified clinical outcomes

The rate of the combined endpoint of death, stroke, systemic emboli and myocardial infarction was 3.6 per 100 person-years (95% CI 3.1–5.1) in actively monitored individuals and 4.5 (95% CI 4.0–5.0) in the observational cohort (adjusted Hazard Ratio 0.79, p = 0.02). (Fig 3) The combined endpoint, excluding death, was also significantly lower, as were the individual endpoints of mortality and stroke (Table 2).

Among all study participants, those who received an AF diagnosis experienced an incident rate of the primary combined endpoint greater than double that of the overall study population

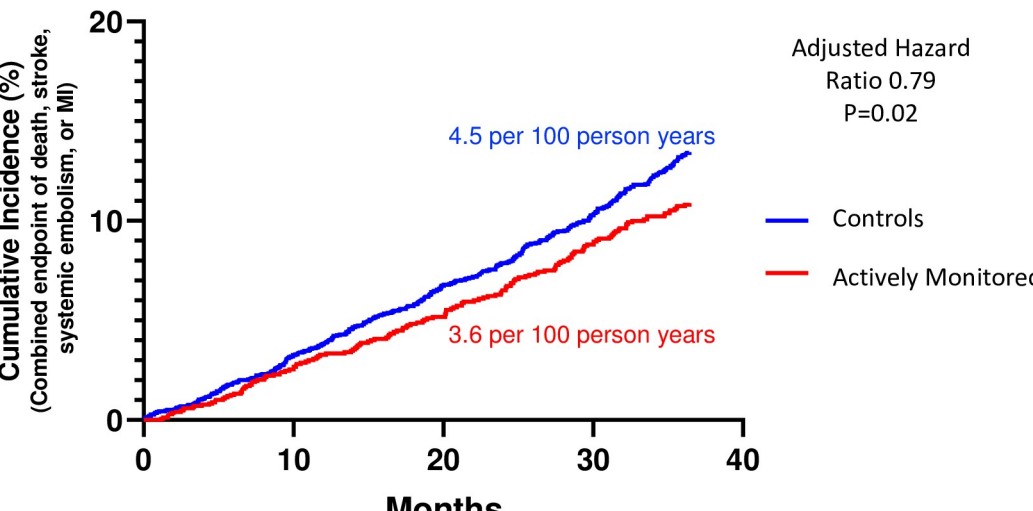

**Fig 3. Primary endpoint.** Cumulative incidence of the combined primary endpoint of death, stroke, systemic emboli or myocardial infarction in actively monitored and observational control cohorts over the 3 years following initiation of screening.

**Table 2. Unadjusted and adjusted clinical outcomes at 3 Years.**

| | Actively Monitored (per 100 person-years (95% CI)) | Observational Control (per 100 person-years (95% CI)) | Unadjusted Hazard Ratio (95% CI) | p-value | Adjusted* Hazard Ratio (95% CI) | p-value |
|---|---|---|---|---|---|---|
| Number | n = 1718 | n = 3371 | | | | |
| Mean follow-up (months) | 30.7 | 28.2 | | | | |
| Stroke | 1.7 (1.3, 2.1) | 2.2 (1.9, 2.6) | 0.74 (0.57, 0.97) | 0.03 | 0.75 (0.57, 0.99) | 0.04 |
| Myocardial Infarction | 1.4 (1.1, 1.8) | 1.6 (1.4, 1.9) | 0.84 (0.62, 1.14) | 0.27 | 0.83 (0.61, 1.13) | 0.22 |
| Systemic Emboli | 0.43 (0.28, 0.68) | 0.57 (0.42, 0.76) | 0.76 (0.44, 1.30) | 0.31 | 0.77 (0.45, 1.32) | 0.34 |
| Death | 0.50 (0.33, 0.76) | 0.81 (0.63, 1.0) | 0.60 (0.37, 0.98) | 0.04 | 0.61 (0.37, 0.99) | 0.047 |
| Stroke, MI, or Emboli | 3.3 (2.8, 3.8) | 4.0 (3.6, 4.5) | 0.81 (0.66, 0.98) | 0.03 | 0.81 (0.66, 0.98) | 0.03 |
| Stroke, MI, Emboli or death | 3.6 (3.1, 4.1) | 4.5 (4.0, 5.0) | 0.80 (0.66, 0.96) | 0.02 | 0.79 (0.66, 0.96) | 0.02 |

* Models adjust for the following baseline (pre-randomization date) covariates: age, female, Charlson Comorbidity Index, heart failure, COPD, chronic renal failure, diabetes, hypertension, obesity, stroke, prior myocardial infarction, sleep apnea, baseline ER visits, baseline PCP visits, and baseline hospitalizations.

(10.6 per 100 person-years), that was significantly lower in the monitored cohort relative to controls (7.1 versus 13.2 per 100 person years, adjusted Hazard Ratio 0.48, p<0.01) (Fig 4A).

As an exploratory analysis, outcomes in individuals in the actively monitored cohort who had no AF during patch monitoring and were subsequently diagnosed clinically were compared to outcomes in the observational cohort, who were similarly diagnosed clinically. The incidence of the combined endpoint of death, stroke, systemic emboli and myocardial infarction per 100 person-years was not statistically different (9.0 versus 13.2, p = 0.06), whereas in those diagnosed by the patch first the incidence was significantly lower than both (2.6, p<0.05 for both) (Fig 4B).

## Events surrounding the new diagnosis of AF

In order to explore the association of clinical events that occurred coincident with a new AF diagnosis we carried out an analysis to identify primary outcome events, as well as a hospitalization and a new diagnosis of heart failure, that occurred in the 4 weeks surrounding their clinical diagnosis of AF. Over half (51.3%) of the individuals in the observational cohort who had a new AF diagnosis had some clinical event (stroke, MI, systemic emboli, a new heart failure diagnosis or hospitalization) in the 4 weeks surrounding that new diagnosis (Fig 5).

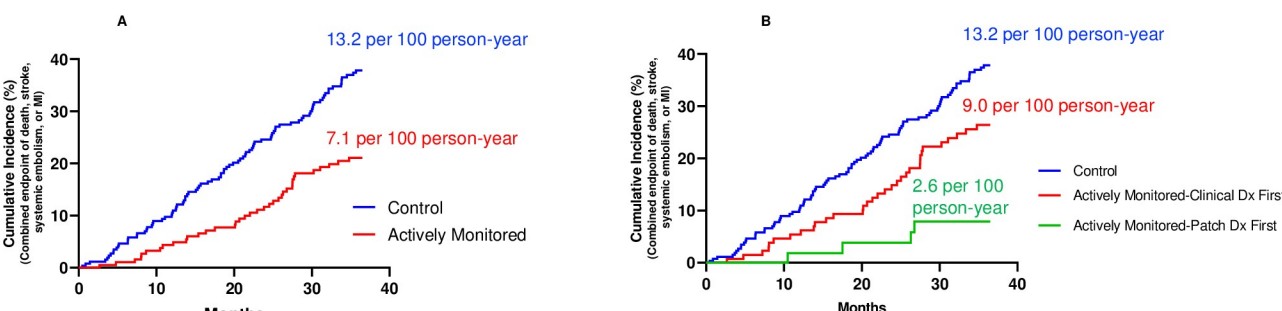

**Fig 4. Primary endpoint in those receiving a new diagnosis of atrial fibrillation.** Cumulative incidence of the combined primary endpoint in individuals diagnosed with atrial fibrillation in A.) the actively monitored and observational control cohorts, and B.) the actively monitored cohort based on whether their initial diagnosis of AF was via ECG patch screening or via a clinical diagnosis, and the observational control cohort.

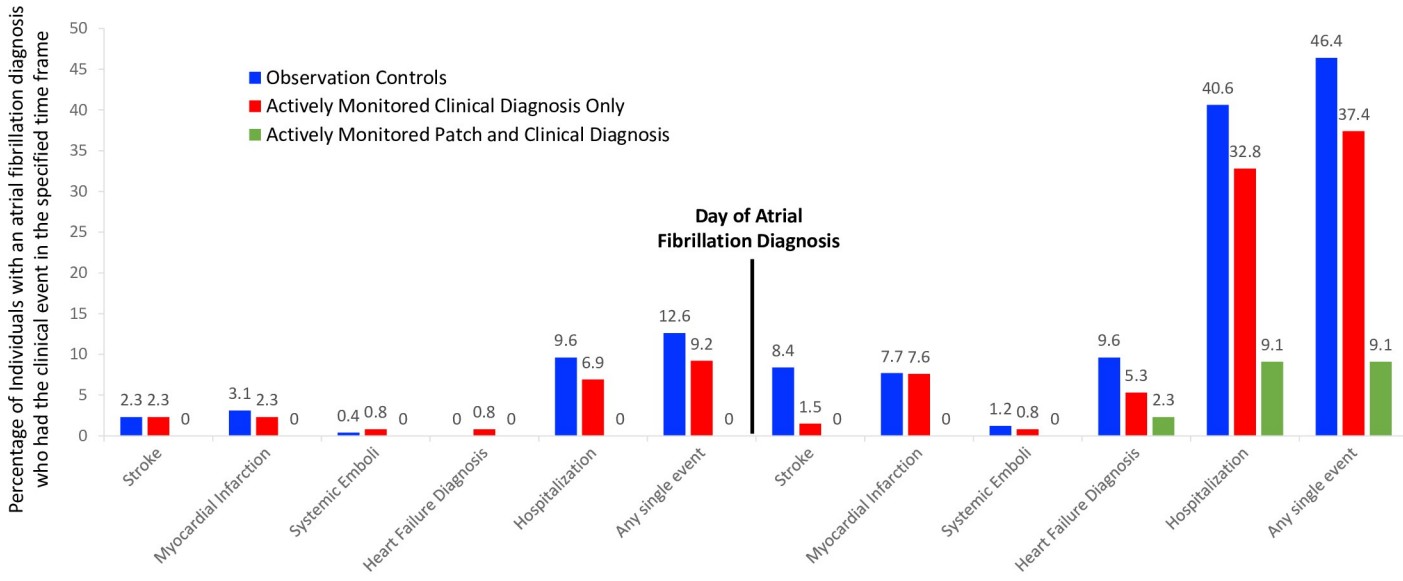

**Fig 5. Clinical events surrounding a new atrial fibrillation diagnosis.** Clinical Events in the two weeks preceding and the 2 weeks following a new atrial fibrillation (AF) diagnosis, inclusive of the diagnosis date, in the observational cohort and the actively monitored cohort based on whether their initial diagnosis of AF was via ECG patch screening or via a clinical diagnosis.

Rates were similar for individuals in the actively monitored cohort whose AF was diagnosed only clinically (42.0%) but was just 9.1% of individuals with patch-diagnosed AF. Of all individuals diagnosed with AF clinically, 6.6% experienced a stroke in the 4 weeks surrounding that event, while 10.2% were newly diagnosed with heart failure. In addition, 9.2% had a myocardial infarction, 1.5% systemic emboli, and cumulatively, 42.9% were hospitalized. For those diagnosed via screening, none experienced a stroke, myocardial infarction or systemic emboli in the period surrounding their AF diagnosis, and only 1 person (2.3%) had a new diagnosis of heart failure.

## Safety endpoint

Rates of hospitalizations for bleeding were 0.32 per 100 person-years in the actively monitored cohort versus 0.71 per 100 person-years in the control cohort with an adjusted Incidence Rate Ratio of 0.47, p<0.01.

## Discussion

Our results add to the limited data currently available describing the impact of ECG-based screening program for undiagnosed AF on clinical outcomes. While we found a significant difference in prespecified clinical outcomes, the lack of randomization and the difference in AF rates at 3 years prevents any strong conclusions about the specific benefit of identifying asymptomatic AF through ECG patch screening to be made from these results. However, our findings still provide valuable, novel information regarding the potential value of active AF screening. We found a surprisingly high rate of clinical events surrounding new, clinically diagnosed AF with 42.9% being hospitalized in the weeks before and after their new AF diagnosis, and with 24.0% receiving a new diagnosis of an irreversible event including stroke, myocardial infarction, systemic emboli or heart failure. On the other hand, people diagnosed with

AF through ECG patch screening not only had a very low rate of clinical events around the time of diagnosis but also throughout the 3-year follow-up period.

We had hypothesized that monitoring with an ECG patch would diagnosis AF earlier, prior to a clinical event and potentially serious complication. We assumed at 3 years that AF rates would be similar between the actively monitored and observational cohorts. This proved not to be the case, making any comparison limited to those with an AF diagnosis purely exploratory. Study results available after mSToPS had completed enrollment support a longer lag time for progression from asymptomatic AF recognized through monitoring to clinically detected AF. Among 415 participants in ASSERT (Asymptomatic Atrial Fibrillation and Stroke Evaluation in Pacemaker Patients and the Atrial Fibrillation Reduction Atrial Pacing Trial) who were found to have an episode of subclinical AF of >6 min but ≤24 hours in duration during the first year after enrollment, only 10.8% (45 of 415) progressed to having clinical AF in the following 2 years [17]. Comparisons between the two full cohorts is also limited since the actively monitored cohort was composed of individuals who all agreed to enroll in a study, and were therefore more likely to be a more activated and health-conscious population than the observational control cohort, which could contribute to better clinical outcomes. In addition, although a lower rate of hospitalization for bleeding in the actively monitored cohort might also suggest they were overall healthier, this could also be related to a clinical diagnosis of AF not detected by screening, and subsequent initiation of anticoagulation, being more likely to occur in the ER or hospital, which has been previously shown to be associated with significantly more hospitalizations for bleeding in the following year [18].

Our primary results, plus those of another recent, randomized study of AF screening using the same long-term ECG patch confirm the ability to identify a substantial minority of at-risk people with undiagnosed AF [12, 19]. We also previously reported that AF screening had no detrimental impact on healthcare resource utilization in the year following screening beyond an increase in cardiology outpatient visits, which were primarily for AF [13]. Recently, the 5-year results from the Systematic ECG Screening for Atrial Fibrillation Among 75-Year-Old Subjects in the Region of Stockholm and Halland, Sweden (STROKESTOP Study) were reported, finding a small but significant decrease in risk of the combined primary endpoint in the group invited for screening [20]. In totality, these findings support the efficacy of ECG screening, the lack of a negative impact on healthcare resource utilization, and its value in improving clinical outcomes, although there is still much work to do to better refine every step of the screening, diagnosis and treatment processes.

Refined screening programs can be designed to take advantage of the growing collection of user-friendly digital health technologies that have enabled the development of multiple large-scale population screening studies involving 100,000's of participants across the globe. Some of these studies require just a one-time ECG, others intermittent, 30-second checks on a recurring basis, several with ECG patches, and still others with wrist-based photoplethysmography sensors [21]. Each of these technologies offers different advantages and disadvantages. For example, patch monitoring, as done in the current study, would be expected to identify roughly twice the number of new AF cases as twice-daily 30 second ECG checks for 14 days as done in STROKESTOP, although the intermittent monitoring would identify a population with a higher burden of AF [22]. The results of the many ongoing studies will add considerable knowledge to the field and accelerate real-world implementation of programs that might prevent significant morbidity and mortality.

Our findings, and those of others, can help inform future screening programs. mSToPS was designed to be pragmatic to better inform clinical implementation at scale, but was not intended to dictate that implementation. As such, we purposely included individuals of moderate risk for AF (e.g. age as low as 55 years) rather than only higher-risk individuals. Future

programs could go beyond clinical factors for risk identification and incorporate genetic, ECG, electronic health record data or all of these and more to better stratify risk [1, 23, 24]. We also did not recommend any specific therapies after a diagnosis of AF but rather left any treatment decision up to the participant and their healthcare provider, which likely negatively impacted the initiation of anticoagulant therapy in qualified participants. Forthcoming screening programs could improve the initiation and maintenance of evidence-based therapies with the incorporation of an app-based tool [25]. In addition, the timing and frequency of ECG patch monitoring were also selected somewhat arbitrarily, leading to only a third of participants eventually diagnosed with AF being diagnosed preclinically via screening, suggesting that recurrent monitoring may be substantially more beneficial.

## Limitations

There are several important limitations to our study as previously described. Beyond the strong possibility of unmeasured confounders, all endpoints were based on claims and membership data, which, although reflective of real-world practice, limited clinical follow-up to the duration of health plan enrollment, which was < 3 years for some participants. In addition, claims data have been found to not be as accurate as physician adjudication [26]. Finally, as previously discussed [12], real world implementation will need to address the fact that approximately one-third of individuals who enrolled in the study never wore their ECG patch monitor.

## Conclusions

Individuals undergoing active screening for AF, as part of a prospective, pragmatic, direct-to-participant, nationwide study, experienced a lower rate of clinical events at 3 years following the initiation of ECG-patch screening relative to routine care, although the impact of earlier diagnosis of AF via screening on this finding is unclear. These observational data support the need for randomized trials to determine whether screening for AF will yield a meaningful protection from strokes and other clinical events.

## Supporting information

**S1 Checklist. CONSORT 2010 checklist of information to include when reporting a randomised trial**[*].
(DOC)

**S1 Data.**
(XLSX)

## Acknowledgments

The authors wish to express their deep gratitude to our partners, the participants of the mSToPS trial, for their kindness and willingness to sacrifice their time to help improve health care.

## Author Contributions

**Conceptualization:** Steven R. Steinhubl, Lauren M. Ariniello, Gail S. Ebner, Katie Baca-Motes, Troy Sarich, Eric J. Topol.

**Data curation:** Steven R. Steinhubl, Jill Waalen, Anirudh Sanyal, Alison M. Edwards.

**Formal analysis:** Jill Waalen, Anirudh Sanyal, Alison M. Edwards.

**Funding acquisition:** Steven R. Steinhubl, Robert A. Zambon, Troy Sarich, Eric J. Topol.

**Investigation:** Steven R. Steinhubl, Lauren M. Ariniello, Gail S. Ebner, Katie Baca-Motes.

**Methodology:** Steven R. Steinhubl, Lauren M. Ariniello, Gail S. Ebner, Katie Baca-Motes, Robert A. Zambon, Troy Sarich, Eric J. Topol.

**Project administration:** Steven R. Steinhubl, Lauren M. Ariniello, Gail S. Ebner.

**Supervision:** Steven R. Steinhubl.

**Validation:** Jill Waalen, Anirudh Sanyal.

**Visualization:** Jill Waalen.

**Writing – original draft:** Steven R. Steinhubl, Jill Waalen, Anirudh Sanyal.

**Writing – review & editing:** Steven R. Steinhubl, Jill Waalen, Anirudh Sanyal, Alison M. Edwards, Lauren M. Ariniello, Gail S. Ebner, Katie Baca-Motes, Robert A. Zambon, Troy Sarich, Eric J. Topol.

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
