## [Decision Letter · Decision Letter 0]

28 Jul 2021

PONE-D-21-22723

3-Year Clinical Outcomes in a Nationwide, Observational, Siteless Clinical Trial of Atrial Fibrillation Screening - mHealth Screening to Prevent Strokes (mSToPS)

PLOS ONE

Dear Dr. Steinhubl,

Thank you for submitting your manuscript to PLOS ONE. After careful consideration, we feel that it has merit but does not fully meet PLOS ONE’s publication criteria as it currently stands. Therefore, we invite you to submit a revised version of the manuscript that addresses the points raised during the review process.

While this Editor much appreciates the Authors' efforts in addressing all Reviewers' comments arisen in the previous round of review at PLOS Medicine, some minor comments have been outlined by the statistical Reviewer in this round of review. However, PLOS One publication criteria appear to have been appropriately met.

We look forward to receiving your revised manuscript.

Kind regards,

Giuseppe Andò, M.D., Ph.D.

Academic Editor

PLOS ONE

Journal Requirements:

"Steve Steinhubl – grant support, Qualcomm Foundation.

Jill Waalen – No conflicts of interest

Anirudh Sanyal - Employee, Healthagen, LLC

Alison Edwards – Employee, Healthagen, LLC

Lauren Ariniello - No conflicts of interest

Gail Ebner - No conflicts of interest

Katie Baca-Motes - No conflicts of interest

Bob Zambon- Employee, Janssen Research & Development; Stockholder Johnson &

Johnson.

Troy Sarich – Employee, Johnson & Johnson; Stockholder Johnson & Johnson.

Eric Topol – grant support, Qualcomm Foundation" 

We note that you received funding from a commercial source: "Qualcomm Foundation, Healthagen, LLC,  Janssen Research & Development and Johnson & Johnson"

please provide an amended Competing Interests Statement that explicitly states this commercial funder, along with any other relevant declarations relating to employment, consultancy, patents, products in development, marketed products, etc. 

"Supported by a research grant from Janssen Pharmaceuticals. Additional support provided

through the National Institutes of Health (NIH)/National Center for Advancing Translational

Sciences grant UL1TR002550 and a grant from the Qualcomm Foundation. "

" Janssen Pharmaceuticals (SRS, GE, LA, KBM, AE, AS). https://www.janssen.com/us/ 

National Institutes of Health (NIH)/National Center for Advancing Translational Sciences  UL1TR002550 (SRS, EJT, GE, LA) https://ncats.nih.gov

 Qualcomm Foundation (SRS, EJT, GE).

The primary funder collaborated in the design and conduct of the study; preparation, review, or approval of the manuscript."

4. Please upload a copy of Supporting Information Table which you refer to in your text on page 4.

Reviewers' comments:

Reviewer's Responses to Questions

**Comments to the Author**

1. Is the manuscript technically sound, and do the data support the conclusions?

Reviewer #1: Yes

2. Has the statistical analysis been performed appropriately and rigorously? 

Reviewer #1: Yes

3. Have the authors made all data underlying the findings in their manuscript fully available?

Reviewer #1: Yes

4. Is the manuscript presented in an intelligible fashion and written in standard English?

Reviewer #1: Yes

5. Review Comments to the Author

Reviewer #1: Interesting and pragmatic study

Abstract: it should be added how screening for AF was performed

42.9% of patients hospitalized for AF: this appears high and should be commented

Methods. Sample size calculation should be better performed. Quotations of incidence of AF should be added

6. PLOS authors have the option to publish the peer review history of their article (what does this mean?). If published, this will include your full peer review and any attached files.

Reviewer #1: **Yes: **Fabrizio D'Ascenzo

---

## [Author Response · Author response to Decision Letter 0]

4 Aug 2021

Response to Reviewers has been uploaded, as requested, as a separate file labeled 'Response to Reviewers'

---

## [Editor Report · Decision Letter 1]

23 Sep 2021

Three Year Clinical Outcomes in a Nationwide, Observational, Siteless Clinical Trial of Atrial Fibrillation Screening - mHealth Screening to Prevent Strokes (mSToPS)

PONE-D-21-22723R1

Dear Dr. Steinhubl,

We’re pleased to inform you that your manuscript has been judged scientifically suitable for publication and will be formally accepted for publication once it meets all outstanding technical requirements.

Kind regards,

Giuseppe Andò, M.D., Ph.D.

Academic Editor

PLOS ONE
---

## [Editor Report · Acceptance letter]

27 Sep 2021

PONE-D-21-22723R1 

Three year clinical outcomes in a nationwide, observational, siteless clinical trial of atrial fibrillation screening - mHealth Screening to Prevent Strokes (mSToPS) 

Dear Dr. Steinhubl:

I'm pleased to inform you that your manuscript has been deemed suitable for publication in PLOS ONE. Congratulations! Your manuscript is now with our production department. 

Kind regards, 

on behalf of

Dr. Giuseppe Andò 

Academic Editor

PLOS ONE